# BUBR1 as a Prognostic Biomarker in Canine Oral Squamous Cell Carcinoma

**DOI:** 10.3390/ani12223082

**Published:** 2022-11-09

**Authors:** Leonor Delgado, Luís Monteiro, Patrícia Silva, Hassan Bousbaa, Fernanda Garcez, João Silva, Paula Brilhante-Simões, Isabel Pires, Justina Prada

**Affiliations:** 1UNIPRO—Oral Pathology and Rehabilitation Research Unit, University Institute of Health Sciences—CESPU (IUCS-CESPU), 4585-116 Gandra, Portugal; 2Pathology Department, INNO Serviços Especializados em Veterinária, 4710-503 Braga, Portugal; 3Medicine and Oral Surgery Department, University Institute of Health Sciences—CESPU (IUCS-CESPU), 4585-116 Gandra, Portugal; 4TOXRUN—Toxicology Research Unit, University Institute of Health Sciences CESPU, CRL, 4585-116 Gandra, Portugal; 5Department of Veterinary Science of the University of Trás-os-Montes and Alto Douro, 5000-801 Vila Real, Portugal; 6CECAV—Veterinary and Animal Research Center, University of Trás-os-Montes and Alto Douro, 5001-801 Vila Real, Portugal

**Keywords:** oral cancer, BUBR1, BUB3, SPINDLY, Ki-67, immunohistochemistry, prognostic markers, survival

## Abstract

**Simple Summary:**

Spindle assembly checkpoint (SAC) includes several proteins that can be dysregulated contributing to oral carcinogenesis. We have investigated the role of some SAC components (BUBR1, BUB3 and SPINDLY proteins) in canine oral squamous cell carcinoma (OSCC) by immunohistochemical analysis of 60 canine OSCCs. We observed that all proteins were detected in almost all cases and with a high expression rate in some cases. Furthermore, we found an independent prognostic value for BUBR1, where high BUBR1 expression was associated with a lower survival rate of these canine patients. These results suggest a potential role of BUBR1 as a prognostic biomarker in canine OSCC and should motivate further studies aimed at the role of these SAC proteins not only as biomarkers but also as pharmacological targets in canine OSCC.

**Abstract:**

Chromosomal instability (CIN) plays a key role in the carcinogenesis of several human cancers and can be related to the deregulation of core components of the spindle assembly checkpoint (SAC) including BUBR1 protein kinase. These proteins have been related to tumor development and poor survival rates in human patients with oral squamous cell carcinoma (OSCC). To investigate the expression of the SAC proteins BUBR1, BUB3 and SPINDLY and also Ki-67 in canine OSCC, we performed an immunohistochemical evaluation in 60 canine OSCCs and compared them with clinical and pathological variables. BUBR1, Ki-67, BUB3 and SPINDLY protein expressions were detected in all cases and classified as with a high-expression extent score in 31 (51.7%) cases for BUBR1, 33 (58.9%) cases for BUB3 and 28 (50.9%) cases for SPINDLY. Ki-67 high expression was observed in 14 (25%) cases. An independent prognostic value for BUBR1 was found, where high BUBR1 expression was associated with lower survival (*p* = 0.012). These results indicate that BUBR1 expression is an independent prognostic factor in these tumors, suggesting the potential use for clinical applications as a prognostic biomarker and also as a pharmacological target in canine OSCC.

## 1. Introduction

Oral cancer makes up for around 7% of canine cancers, with oral squamous cell carcinoma (OSCC) being the second most common histological type reported in most of the international literature [1]. They are more commonly seen in large-breed dogs, mostly above 7 years old and with gingiva being one of the most common locations. Nevertheless, scarce information exists on the etiopathogenesis of these neoplasms in dogs in contrast with the well-known role of tobacco consumption and alcohol misuse in human OSCC [2,3,4]. Moreover, the prognosis (including survival studies) of dogs with OSCC is poorly documented and reported in the literature [5,6,7].

Cancer progression, namely, OSCC development, is characterized by multiple genetic and epigenetic events [8]. Understanding the alterations that occur on a molecular level may provide new insights into therapies as well as diagnostic tools for earlier and more effective diagnosis of OSCC in dogs.

The existence of chromosomal instability (CIN) is an important step for tumorigenesis and has been reported in human OSCC [9,10,11]. The spindle assembly checkpoint (SAC) is a protective cell device phase that controls and delays the mitotic progression until all chromosomes establish proper attachments to spindle microtubules and are correctly aligned in the metaphase. Several proteins participate in this SAC, including BUB1, BUB3, MAD1, MAD2, BUBR1, MPS1 and Aurora B proteins [12]. The main event in the SAC is the activation of the mitotic checkpoint complex (MCC): an inhibitory signal of the metaphase to anaphase transition. This complex is composed of several proteins, MAD2, BUBR1, and BUB3 in association with CDC20 protein, which, when activated, blocks the binding of CDC20 to the anaphase-promoting complex/cyclosome (APC/C), preventing anaphase onset. BUBR1 has an important role in SAC signaling and is also involved in the establishment of proper kinetochore microtubule attachments (Figure 1). Their inhibition results in severe chromosome segregation defects [13,14,15,16,17]. Another important regulator of chromosome alignment and SAC signaling is the protein SPINDLY which acts with dynein protein during the mitosis process [18].

Some studies reported that defects in SAC activity are associated with an increased rate of aneuploidy development during tumorigenesis and that this is related to abnormal expression levels of SAC components [11,19]. However, to our knowledge, there is no information concerning SAC component expression in canine oral cancer, and their role in tumor progression and survival of dogs with these tumors is unknown.

The aim of the present study is to evaluate the immunohistochemical expression of SAC components, namely, BUBR1, BUB3, SPINDLY and also the proliferative marker Ki-67 in canine OSCC, and relate them to the clinical, pathological and prognostic features in a cohort of dog patients.

## 2. Materials and Methods

### 2.1. Patients and Tissue Specimens

We included 60 cases of canine OSCC from the Pathology Laboratory—INNO—(formalin-fixed and paraffin-embedded tissue samples) that were submitted by Portuguese veterinary hospitals and clinics and diagnosed between 1 January 2010 and 31 December 2017. The ethical principles of animal research were followed, and the study was approved by the INNO lab administration board (nº INNO/2021/01).

We included cases with tumors located in the oral cavity (ICD 10: C00-06) and with a confirmed histopathology diagnosis of OSCC. Cases with a previous history of oral cancer treatment or cases without any histopathological confirmation were excluded.

The variables analyzed comprised age, gender, breed, localization of the lesion, size of the lesion and tumor stage [20,21], histopathological diagnosis [22], histopathological grade according to Anneroth et al. (1987) [23] and Bryne’s classifications [24], presence of bone and vascular invasion, number of mitoses, presence of necrosis, presence of lymphocytic infiltration and survival follow-up. We considered the follow-up period obtained through information on the available patient data of the clinical units. The clinical state of the patients was categorized as alive without oral cancer, alive with oral cancer, died from oral cancer (including euthanized patients with oral cancer) and died from other causes.

As a result of the inclusion and exclusion criteria, the total sample was composed of 60 cases of canine OSCC with 32 males (53.3%) and 28 females (46.7%) (ratio 3:1), and ages ranging from 1 to 15 years, with an average age of 10.90 ± 2.74 years. Table 1 presents further clinical and pathological characteristics.

### 2.2. Immunohistochemistry Processing and Evaluation

The expression of BUBR1, BUB3, SPINDLY and Ki-67 proteins was evaluated through immunohistochemistry on silane-coated 3 µm tissue sections. Firstly, the samples were dewaxed in xylene and hydrated through a decreasing series of alcohol concentrations, followed by antigen-retrieval treatment (EDTA buffer 0.01 M pH 9.0 for BUBR1 and Ki-67 and citrate buffer pH 6.0 for BUB3 and SPINDLY) at a high temperature (water bath, 30 min at 98 °C).

After blocking for nonspecific binding (using 3–4% (*v*/*v*) hydrogen peroxide and 0.4% casein in phosphate-buffered saline), the primary antibody was added to the sections and incubated for 60 min at room temperature. The following primary antibodies were used: mouse anti-human BUBR1 diluted at 1:150 (clone 9, BD Biosciences, Sparks, MD, USA); mouse anti-human Ki-67 diluted at 1:100 (clone MIB1, Dako, Glostrup, Denmark); rabbit anti-human BUB3 diluted at 1:500 (clone EPR5319(2), ab133699, Abcam, Cambridge, UK); and rabbit anti-human SPINDLY diluted at 1:500 (HPA044700, Sigma-Aldrich, Darmstadt, Germany). The cross-reactivity of the antibodies between human and canine species was previously evaluated and based on amino acid sequence conservation of the peptide recognized by the antibodies, as shown in Appendix A and Appendix A. A standard peroxidase-labeled dextran polymer was used for visualization with diaminobenzidine as chromogen (NovoLink Polymer Detection System; Novocastra, Leica Biosystems Newcastle, UK), according to the manufacturer’s instructions. Then, sections were lightly counterstained with Gill’s hematoxylin and cover slipped. Positive (oral mucosa and normal testis tissue, both human and canine) and negative controls (omission of primary antibody) were included in each staining run [11,16].

BUBR1, BUB3, SPINDLY and Ki-67 immunoreactivity was assessed semiquantitatively in 10 high-power fields (considering a minimum count of 150 tumor cells per field), and, based on the extent of labeling, scored as: 0, absent; 1, 1–9% of tumor cells labeled; 2, 10–24% of tumor cells labeled; 3, 25–49% of tumor cells labeled; 4, 50–74% of tumor cells labeled; and 5, 75–100% of tumor cells labeled. The intensity of staining was assessed as 0 (negative), 1 (weak), 2 (moderate) and 3 (strong). For data analysis, the labeling index of ≤49% or >50% of labeled tumor cells was used for BUBR1, SPINDLY and Ki-67, and the labeling index of ≤74% or >75% was used for BUB3, as previously described [16]. For all markers, the intensity of staining was considered as low for scores ≤2 and as high for scores of 3. A ZEISS AxioLab A1^®^ microscope (Carl Zeiss Microscopy GmbH, Jena, Germany) with a ZEISS Axiocam 105 color ^®^ and ZEISS Zen2^®^ software (Carl Zeiss Microscopy GmbH, Jena, Germany) was used by two observers (LD and LM) for independently evaluating all samples. The discordant cases were reviewed by both observers together to achieve a consensus.

### 2.3. Statistical Analysis

The software IBM SPSS Statistics version 27.0 (IBM Corporation, Armonk, NY, USA) was used for statistical analysis. Possible associations between the variables were analyzed using the Mann–Whitney test (2 samples) or Kruskal–Wallis one-way ANOVA test with pairwise multiple comparisons (with Bonferroni adjustment where applicable). The correlation between BUBR1, Ki-67, BUB3 and SPINDLY was measured with Spearman’s correlation coefficient.

Cancer-specific survival (CSS) was defined as the time interval (months) between histologic diagnosis and death as a result of oral cancer (including euthanized patients with oral cancer). Survival times of the patients who were still alive or who died from causes not associated with oral cancer were considered for CSS using the last date that they were seen alive or the date of death, respectively.

For univariate survival analysis, Kaplan–Meier and log-rank tests were performed. For multivariable analysis, Cox proportional hazards model was used (using the *Enter method* of Cox regression), considering variables with significant influence in the univariate analysis.

For all tests, the level of significance was set at probabilities of *p* < 0.05.

## 3. Results

### 3.1. BUBR1

The BUBR1 expression was detected in all cases and classified into the following categories: 0–9% in 2 cases (3.3%), 10–24% in 13 (21.7%), 25–49% in 14 (23.3%), 50–74% in 25 (41.7%) and 75–100% in 6 cases (10%). For data analysis, the extent of BUBR1 expression was divided into low expression (0–49%) in 29 (48.3%) cases and high expression in 31 (51.7%) cases (Figure 2). Regarding the intensity of the biomarker, 5 cases (8.3%) showed a weak intensity, 25 (41.7%) a moderate intensity and 30 cases (50%) a strong intensity of staining. These results were grouped into weak and moderate intensity versus strong intensity for statistical analysis.

In addition to the tumor cell cytoplasm, the biomarker was also detected in the tumor nucleus in 27 (45%) of the cases. Analyzing the distribution pattern, it was observed mainly in the periphery cells of tumor islands (N = 41; 68.3%), or homogeneously distributed throughout the tumor island in 14 (23.3%) cases and multifocal in 5 cases (8.3%).

In the analysis of the association between BUBR1 and the clinical-pathological characteristics, a significant association was observed between a high expression (extent score) of BUBR1 and breed (*p* = 0.03) (higher in large breeds), histological type (*p* = 0.029) (higher in conventional SCC), presence of bone invasion (*p* = 0.029), stage of invasion (*p* = 0.029) (higher in stage III) and advanced tumor stage (*p* = 0.034). In addition, an association was observed between a high-intensity BUBR1 score and the presence of bone invasion (*p* = 0.039) (Appendix A).

### 3.2. BUB3

The BUB3 was present in all cases, appearing in the nucleus of tumor cells, although staining was detected also in the tumor cytoplasm of four cases (8.8%). The extent of the biomarker was classified as: 0–9% in one case (1.8%), 10–24% in another case (1.8%), 25–49% in four (7.1%) cases, 50–74% in seventeen (30.4%) cases, and 75–100% in thirty-three (58.9%) cases. For data analysis, BUB3 expression was classified as low expression in 23 (41.1%) cases and high expression in 33 (58.9%) cases (Figure 3).

The intensity score corresponded to weak in 7 cases (12.5%), moderate in 31 cases (55.4%) and strong in 18 (32.1%) cases. We observed that most of the detected tumor cell staining was distributed homogenously within the tumor islands (48; 85.7%).

No significant associations were observed between BUB3 and clinical-pathological characteristics (Appendix A).

### 3.3. SPINDLY

SPINDLY expression was found in all cases and observed in the cytoplasm of tumor cells classified into the categories of 0–9% in 8 (14.5%) cases, 10–24% in 9 (16.4%), 25–49% in 10 (18.2%), 50–74% in 17 (30.9%) and 75–100% in 11 (20%) cases. For data analysis, SPINDLY expression was grouped into low expression in 27 (49.1%) cases and high expression in 28 (50.9%) cases (Figure 4). Intensity staining was recorded as weak in 19 (34.5%) cases, moderate in 30 (54.5%) and strong in 6 cases (10.9%).

Most of the cases (n = 43; 78.2%) presented a heterogenous (periphery/or multifocal) distribution of staining, and 12 cases (21.8%) presented a homogeneous distribution.

A significant association was observed between a high expression (extent score) of SPINDLY and the number of mitoses (*p* = 0.037) (higher expression in 0–1 mitoses/HPF) and lymphocytic infiltration (*p* = 0.046) (higher expression in lymphocytic infiltration weak group) (Appendix A).

### 3.4. Ki-67

Ki-67 nuclear expression was detected in all cases, classified as 0–9% in 3 (5.4%) cases, 10–24% in 18 (32.1%), 25–49% in 21 (37.5%), 50–74% in 12 (21.4%) and 75–100% in 2 (3.6%) cases. For data analysis, Ki-67 expression was classified as low expression in 42 (75%) cases and high expression in 14 (25%) cases (Figure 5).

We also analyzed the intensity of Ki-67 staining that corresponded to weak in 7 cases (12.5%), moderate in 29 (51.8%) and strong in 20 (35.7% cases). Most of the cases presented staining distributed in the periphery of tumor islands (42; 75%), but in 10 cases (17.8%) this was heterogeneously multifocal, and in 4 cases (7.1%) it was homogeneously stained.

The comparison of Ki-67 and the clinical-pathological characteristics reveals significant associations between a high expression (extent score) of Ki-67 and the pattern of invasion (*p* = 0.007) (higher in stage III) (Appendix A).

### 3.5. Associations between Biomarkers

BUBR1 extent was correlated with SPINDLY (*p* = 0.032) extent and Ki-67 extent (*p* < 0.001), and BUBR1 intensity with SPINDLY intensity (<0.001). Ki-67 extent was also correlated with the SPINDLY extent score (*p* = 0.009) (Table 2 and Figure 6).

### 3.6. Clinical Outcome

Fifty patients had information concerning survival (including dates of the dead when applicable) (mean follow-up of 6.8 ± 1.5 months), of which five (10%) were alive without oral cancer, six (12%) were alive with oral cancer, thirty-three patients (66%) had died as a result of oral cancer and six (12%) died from other causes.

During Kaplan–Meier (and log-rank test) analysis, histological type (*p* = 0.013), pattern of invasion (*p* = 0.011), stage of invasion (*p* = 0.009), type of treatment (*p* = 0.048), tumor stage (*p* = 0.001) and BUBR1 extent (*p* = 0.002) were statistically associated with CSS, as can be observed in Appendix A and Table 3, and also Figure 7.

Taking into account these results, we included the variables with statistically significant results in a multivariate analysis. We observed an independent prognostic value of BUBR1 extension where tumors with a high BUBR1 expression had a lower CSS (HR, 5.462; 95% CI, 1.455–20.502, *p* = 0.012). A palliative or support approach (HR, 6.214; 95% CI, 1.433–26.943, *p* = 0.015), patients with an advanced tumor stage (HR, 5.005; 95% CI, 1.405–17.829, *p* = 0.013), tumors classified with a pattern of invasion III (HR, 5.910; 95% CI, 1.160–30.118, *p* = 0.033) or stage of invasion IV (HR, 15.686; 95% CI, 1.272–193.435, *p* = 0.032) also presented a significant and independent lower CSS compared with reference categories (Table 4).

## 4. Discussion

The comprehension of the expression profile of SAC components may be useful to understand oral cancer development and eventually useful in the development of anti-SAC agents’ therapies for the treatment of these tumors in dogs, similar to the approaches undergoing for the treatment of human malignancies [25,26].

This study provides the first analysis of SAC-related proteins BUBR1, BUB3 and SPINDLY in canine oral squamous cell carcinoma, correlating them with clinical-pathological parameters and a prognosis profile. We found that all cases expressed BUBR1 (with a variable range of staining), with the majority of cases revealing a high expression (51.7%) of the biomarker. To our knowledge there are no other studies evaluating these markers on canine samples available for comparison to the present study, however the obtained values could be compared to previous studies on human OSCC. Teixeira et al. [11] reported BUBR1 labeling of 82.5% of OSCC tissue samples, mainly at the periphery of the tumor islands, as observed in our cases. In other study, BUBR1 was expressed in all 43 (100%) OSCCs, mostly in a cytoplasmic location [27], and was found to be overexpressed in 11 of the 49 cases (22.4%) with the localization mainly confined to cytoplasm in the report of Lira et al. [28]. In another study, BUBR1 high expression was observed in 19/27 (70%) of OSCCs without metastasis and 18/23 (78%) of OSCCs with metastasis [29].

In the present study, we found a significant association between BUBR1 high expression (extent score) and breed, histological type, presence of bone invasion, stage of invasion and tumor stage. In addition, an association was observed between a high-intensity BUBR1 score and the presence of bone invasion. The reason for the association with the breed is not easily understandable, especially without other studies for comparison with this, but the association with the histological type (higher in conventional OSCC) and a more advanced invasion or tumor stage could indicate that this protein could be related to the progression and dissemination of the tumor. In fact, in some human cancers, such as gastric, urothelial bladder, prostatic and oral cavity cancers, the SAC component expression status is linked with cancer progression, a high proliferation activity and a poor prognosis [14,30,31,32]. Nevertheless, not all studies have reported this direct association of this biomarker with these variables, expressing, perhaps, the heterogeneity and variability of OSCC [28].

Concerning BUB3, an immunohistochemical expression was present in most of the cases, with staining homogenously distributed within tumor islands. Most of the tumor cells showed expression of this marker, and differences between cases were mainly observed regarding the intensity of the biomarker. No significant associations were observed between BUB3 and clinical-pathological characteristics. In a previous study on human OSCC, out of 62 OSCCs, BUB3 expression was detected in most of the cases, and no significant correlation was found with the clinicopathologic factors, except for treatment modality [16]. Although this protein is present in OSCC, its value as a biomarker is still too uncertain to be used in these neoplasms.

SPINDLY was observed in the cytoplasm of the tumor cells in many cases and related to the number of mitoses and lymphocytic infiltration. Nevertheless, we observed in our sample that there was an inverse association with the number of mitoses, as a higher number of mitosis/HPF corresponded to a lower SPINDLY expression. This contrasts with other reports, especially in human OSCC [16], and could be related to the present sample constitution with other regulation mechanisms by other genes/proteins or pathways. The lack of previous data on canine OSCC precluded any comparison, stressing the need for further analysis of the role of SPINDLY in canine OSCC.

The analysis of survival in biomarker studies on canine samples has been difficult and scarce. We included information on the clinical state of the patients after initial diagnosis with follow-up information obtained from the clinical units (during contact and by questionnaire). In the univariate analysis, some variables revealed a significant influence on CSS, including treatment, tumor stage, the pattern of invasion or stage of invasion, confirming their influence on the prognosis of these patients. However, from the biomarkers evaluated, only BUBR1 showed a significant relation with CSS in univariate and multivariate analyses. Importantly, it showed an independent prognostic value where tumors with a high BUBR1 expression had a lower CSS (HR, 5.462; 95% CI, 1455–20.502, *p* = 0.012).

The association of BUBR1 to prognostic variables has been reported in human OSCC [11], prostate cancer [32] and hepatocellular carcinomas [33]. Nevertheless, some conflicting results are reported showing the opposite result on survival or recurrence rates [28]. Moreover, in a study on human gastric cancer, the authors observed that overexpression of BUBR1 was associated with a low risk of gastric cancer progression, and that overexpression of BUBR1 can therefore be used to identify gastric cancer patients with a favorable prognosis [34]. This could indicate that the function and the significance of this biomarker are different regarding tumor location. Our results on the present sample not only suggest that BUBR1 could be a useful biomarker for the prognostication of canine OSCC, but also that it seems that there is a similarity regarding SAC pathway between human and canine oral tumorigenesis. This biomarker should also be evaluated as a possible target for directed molecular therapies for these tumors. Interestingly, in oral potentially malignant disorders, such as oral leukoplakia, BUBR1 and MAD2 were associated with an increased risk of malignant transformation independent of histological grade and could be potential and useful predictive risk biomarkers of malignant transformations [35]. It would be interesting in the future to evaluate if this could be applied also to canine oral potential malignant disorders. In addition, the relation of the expression of these proteins with other events (e.g., apoptosis escape, aneuploidy and metabolic regulation) should be evaluated in future research. This also includes the use of other methodologies such as Western blot, mRNA levels via RT-PCR, or ploidy analysis.

Other biomarkers such as Ki-67 or BUB3 showed a trend of being higher in cases with poor survival rates, but without significant results. BUB3 was an independent prognostic indicator for cancer-specific survival in human OSCC [16] and also showed an independent prognostic value in prostate cancer for the dichotomized scores of cytoplasmic BUB3, CCNB1 (cyclin B1) and PTTG1 (pituitary tumor-transforming gene) in both univariable and multivariable analyses. The present results suggest that research should also be directed in the future to the use of these SAC proteins as targets for molecular therapies in these canine tumors.

We recognize potential limitations in this study, many related to its retrospective nature, small amount of cases, relatively short follow-up time, direction toward immunohistochemistry analysis, and lack of some clinical data that contributed to the exclusion of some cases. Nevertheless, by performing multivariate analysis, we managed to control some potentially confounding variables and reported for the first time a promising role of these biomarkers, especially BUBR1, in canine OSCC.

## 5. Conclusions

The present data are the first report of BUBR1 as an independent prognostic factor in canine OSCC. With this work, we contributed to expounding the SAC status in oral canine cancer and suggesting BUBR1 as a potential candidate for clinical applications as an OSCC prognostic factor and also as a pharmacological target.

## Figures and Tables

**Figure 1 animals-12-03082-f001:**
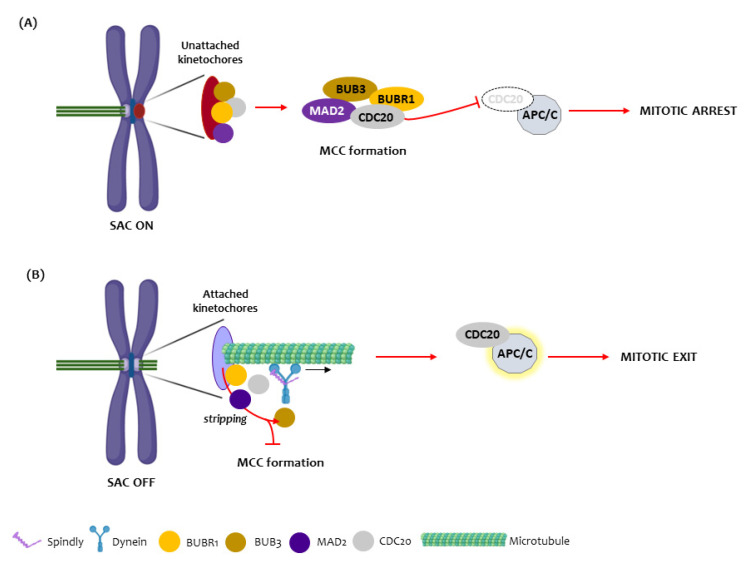
The Spindle assembly checkpoint mechanism. (**A**) In the presence of unattached or misattached kinetochores (red circle), the spindle assembly checkpoint is turned on (SAC ON) and the kinetochore proteins BUB3, BUBR1, MAD2 and CDC20 assemble at the cytosol to form the mitotic checkpoint complex (MCC). Once the MCC is generated, the CDC20 becomes unable to bind to the anaphase-promoting complex/cyclosome (APC/C), leading to mitotic arrest at the metaphase to anaphase transition. (**B**) Upon appropriate kinetochore microtubule attachments, the SAC is turned off (SAC OFF) through dynein/SPINDLY-mediated stripping of BUB3, BUBR1, MAD2 and CDC20 from attached kinetochores, which prevents the assembly of new MCC and through disassembly of existing MCC (not shown). Both mechanisms release CDC20 which, thus, binds to and activates the APC/C, promoting mitosis exit.

**Figure 2 animals-12-03082-f002:**
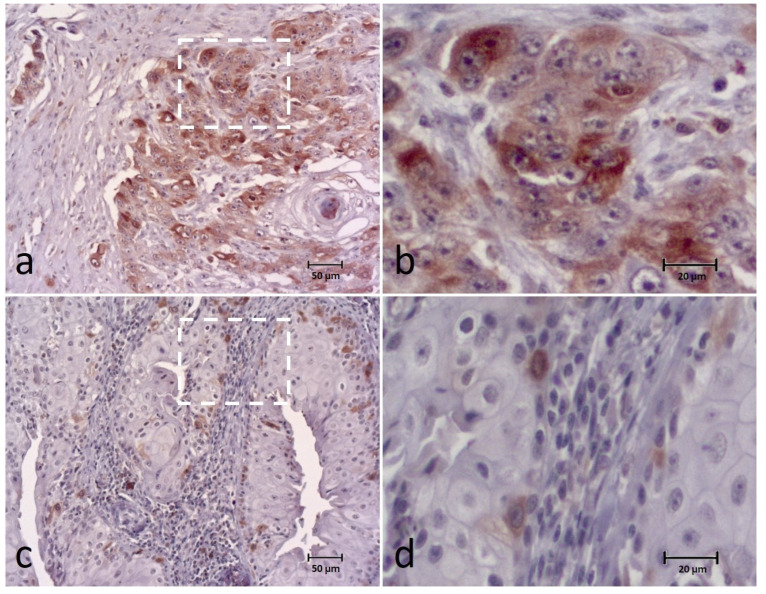
Immunohistochemical (IHC) staining of the BUBR1 in canine OSCC (**a**–**d**) showing high expression (extent and intensity) (**a**,**b**) and low expression (extent and intensity) (**c**,**d**) of the biomarker. Figures (**b**,**d**) correspond to a higher magnification of the white square indicated in cases (**a**,**c**), respectively.

**Figure 3 animals-12-03082-f003:**
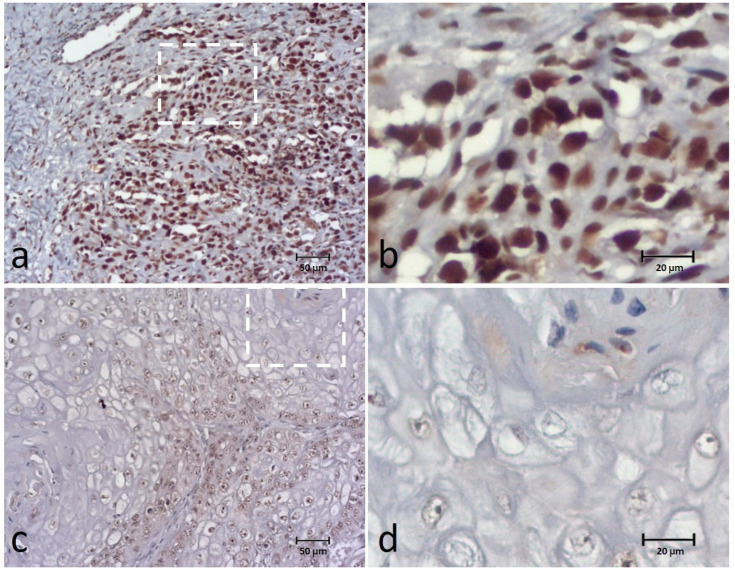
Immunohistochemical (IHC) staining of the BUB3 in canine OSCC (**a**–**d**) showing high expression (extent and intensity) (**a**,**b**) and low expression (extent and intensity) (**c**,**d**) of the biomarker. Figures (**b**,**d**) correspond to a higher magnification of the white square indicated in cases (**a**,**c**), respectively.

**Figure 4 animals-12-03082-f004:**
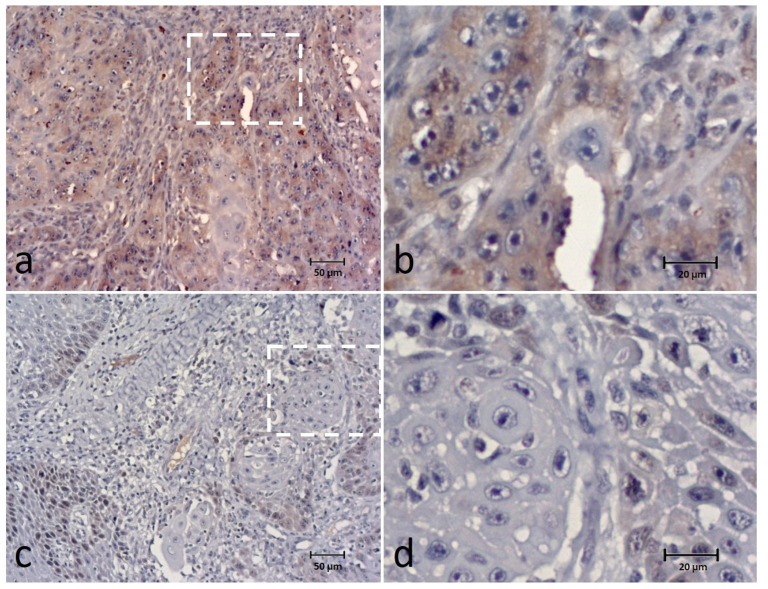
Immunohistochemical (IHC) staining of the SPINDLY in canine OSCC (**a**–**d**) showing high expression (extent and intensity) (**a**,**b**) and low expression (extent and intensity) (**c**,**d**) of the biomarker. Figures (**b**,**d**) correspond to a higher magnification of the white square indicated in cases (**a**,**c**), respectively.

**Figure 5 animals-12-03082-f005:**
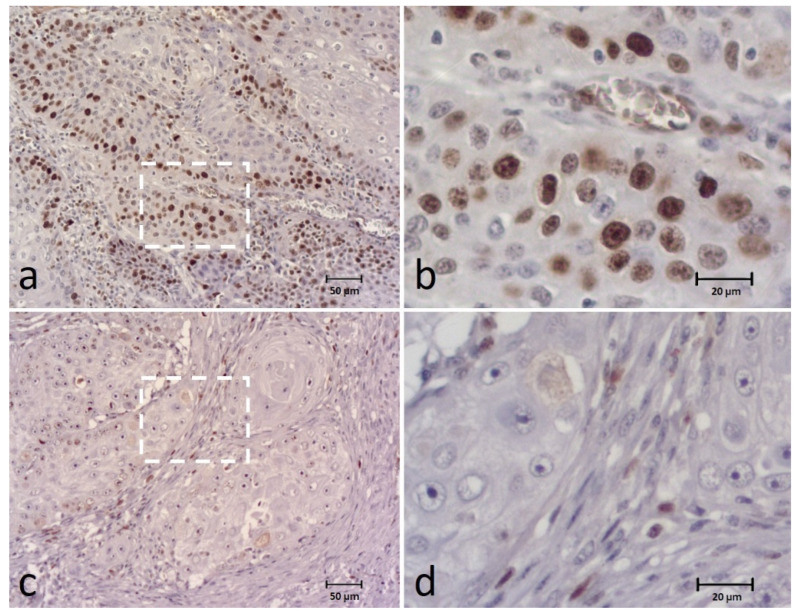
Immunohistochemical (IHC) staining of the Ki-67 in canine OSCC (**a**–**d**) showing high expression (**a**,**b**) and low expression (**c**,**d**) of the biomarker. Figures (**b**,**d**) correspond to a higher magnification of the white square indicated in cases (**a**,**c**), respectively.

**Figure 6 animals-12-03082-f006:**
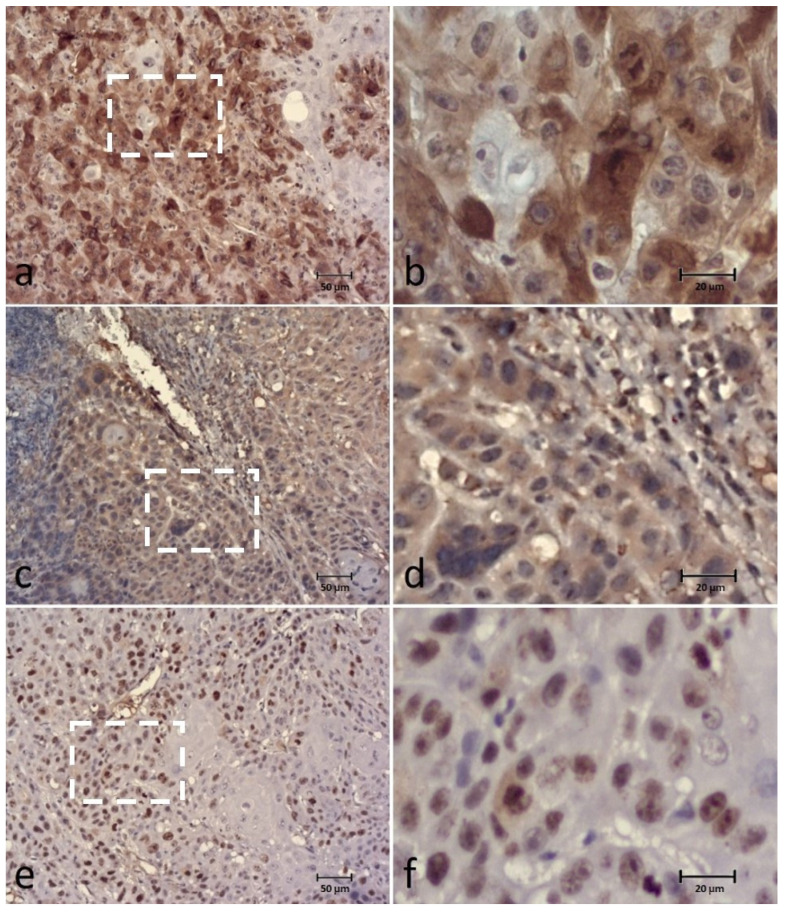
Immunohistochemical (IHC) staining of the BUBR1 (**a**,**b**), SPINDLY (**c**,**d**) and Ki-67 (**e**,**f**) in the same case of canine OSCC, showing high expression of all biomarkers. Figures (**b**,**d**,**f**) correspond to a higher magnification of the white square indicated in cases (**a**,**c**,**e**), respectively.

**Figure 7 animals-12-03082-f007:**
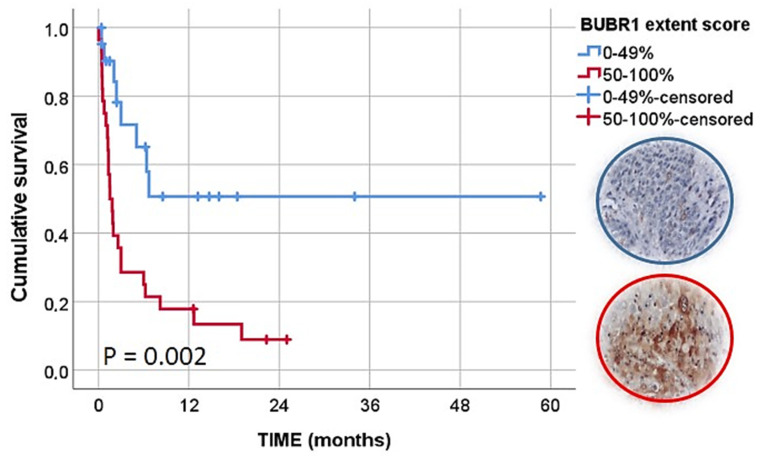
Kaplan–Meier curves of CSS analysis according to BUBR1 extent score.

**Table 1 animals-12-03082-t001:** Patient characteristics (n = 60).

Variables		N (%)
**Gender**	Female	28 (46.7%)
	Male	32 (53.3%)
**Age**	<7 years old	6 (10%)
	≥7 years old	54 (90%)
**Breed** **(* n = 53)**	Small	10 (18.9%)
	Medium	5 (9.4%)
	Large	12 (22.6%)
	UB	26 (49.1%)
**Tumor location**	Mouth (NOS)	13 (21.7%)
	Gingiva	20 (33.3%)
	Tongue	12 (20%)
	Oropharynx (including tonsils)	8 (13.3%)
	Palate	7 (11.7%)
**Histological type**	Papillary SCC	10 (16.7%)
	Conventional SCC	50 (83.3%)
**Anneroth’s histological grade**	Well-differentiated	19 (31.7%)
	Moderately differentiated	41 (68.3%)
	Poorly differentiated	0 (0%)
**Bryne´s histological grade**	Well-differentiated	25 (41.7%)
	Moderately differentiated	33 (55%)
	Poorly differentiated	2 (3.3%)
**Pattern of invasion**	I—Pushing, well-delineated, infiltrating borders	20 (33.3%)
	II—Infiltrating, solid cords, bands and/or strands	22 (36.7%)
	III—Small groups or cords of infiltrating cells	15 (25%)
	IV—Marked and widespread cellular dissemination in small groups and/or in single cells	3 (5%)
**Stage of invasion**	I—Carcinoma in situ and/or questionable invasion	0
	II—Distinct invasion, but involving lamina propria only	39 (65%)
	III—Invasion below lamina propria adjacent to muscles, salivary gland tissues and periosteum	20 (33.3%)
	IV—Extensive and deep invasion replacing most of the stromal tissue and infiltrating jaw bone	1 (1.7%)
**Bone invasion**	Absent	50 (83.3%)
	Present	10 (16.7%)
**Vascular invasion**	Absent	56 (93.3%)
	Present	4 (6.7%)
**Tumor stage** **(* n = 50)**	I + II	19 (38%)
	III + IV	31 (62%)
**Treatment** **(* n = 50)**	Surgery	11 (22%)
	Chemotherapy	4 (8%)
	Palliative treatment/support	35 (70%)

Legend: UB, undetermined breed (including mixed breeds); NOS, not otherwise specified; SSC, squamous cell carcinoma; * information not available for some patients.

**Table 2 animals-12-03082-t002:** Correlation between the biomarkers BUBR1, BUB3, SPINDLY and Ki-67.

Variables		BUBR1		BUB3		SPINDLY		Ki-67	
	Score	Extent	Intensity	Extent	Intensity	Extent	Intensity	Extent	Intensity
**BUBR1**	Extent	-	-		-		-		-
	ρ			0.071		0.290		0.506	
	*p*-value			0.601		**0.032**		**<0.001**	
	Intensity	-	-	-		-		-	
	ρ				0.182		0.452		0.182
	*p*-value				0.180		**<0.001**		0.180
**BUB3**	Extent		-	-	-		-		-
	ρ	0.071				0.089		0.225	
	*p*-value	0.601				0.526		0.102	
	Intensity	-		-	-	-		-	
	ρ		0.182				0.180		0.138
	*p*-value		0.180				0.196		0.320
**SPINDLY**	Extent		-		-	-	-		-
	ρ	0.290		0.089				0.148	
	*p*-value	**0.032**		0.526				0.286	
	Intensity	-		-		-	-	-	
	ρ		0.457		0.180				0.352
	*p*-value		**<0.001**		0.196				**0.009**
**Ki-67**	Extent		-		-		-	-	-
	ρ	0.506		0.225		0.148			
	*p*-value	**<0.001**		0.102		0.286			
	Intensity	-		-		-		-	-
	ρ		0.182		0.138		0.352		
	*p*-value		0.180		0.320		**0.009**		

ρ, *rho*; numbers in bold are considered as a significant *p*-value.

**Table 3 animals-12-03082-t003:** Univariate analysis of cancer-specific survival (CSS) of analyzed biomarkers.

Factors ^†^	Factors	N	Dead	CSS 1 Year *	CSS 2 Years *	CSS Mean CI 95% **	*p*-Value
**BUBR1 extent**	0–49%	22	8	50.7	50.7	31.55 ± 7.02 (17.79–45.31)	0.002
	50–100%	28	25	17.9	8.9	5.32 ± 1.45 (2.49–8.16)	
**BUBR1 intensity**	Negative/weak/moderate	22	12	32	32	20.9 ± 6.152 (8.850–32.97)	0.422
	Strong	28	20	29.2	15.6	9.001 ± 2.525 (4.06–13.96)	
**BUBR1 location**	Cytoplasm	26	15	41.8	33.4	22.35 ± 5.82 (10.94–33.77)	0.213
	Cytoplasm + nucleus	24	18	18.6	0.00	4.79 ± 0.97 (2.89–6.68)	
**BUBR1 distribution**	Homogeneous	12	9	27.9	00.0	7.24 ± 2.09 (3.139–11.34)	0.972
	Periphery	34	22	29.6	29.6	18.87 ± 4.66 (9.37–28.02)	
	Multifocal	4	2	37.5	37.5	5.82 ± 3.37 (0–12.43)	
**Ki-67 extent**	0–49%	35	19	38.5	38.5	24.00 ± 5.08 (14.05–33.95)	0.06
	50–100%	14	13	15.5	0.00	4.81 ± 1.55 (1.77–7.84)	
**Ki-67 intensity**	Negative/weak/moderate	29	18	31.3	23.5	16.53 ± 52.78 (6.19–26.88)	0.707
	Strong	20	14	32.1	24.1	11.47 ± 3.135 (5.32–17.61)	
**Ki-67 distribution**	Homogeneous	4	3	0.00	0.00	1.13 ± 0.27 (0.60–1.67)	0.08
	Periphery	36	23	32.6	26.1	18.35 ± 4.63 (9.27–27.43)	
	Multifocal	9	6	40.0	20.0	6.62 ± 1.87 (2.95–10.29)	
**BUB3 extent**	Low (0–74%)	16	13	13.9	13.9	6.432 ± 2.02 (2.47–10.40)	0.800
	High (75–100%)	33	20	38.2	21.8	16.67 ± 5.30 (6.28–27.07)	
**BUB3 intensity**	Negative/weak/moderate	31	19	30.7	30.7	19.61 ± 5.08 (9.64–29.58)	0.546
	Strong	18	14	26.5	9.9	7.27 ± 2.06 (3.23–11.32)	
**BUB3 distribution**	Homogeneous	44	29	31.2	22.2	16.31 ± 4.04 (8.37–24.24)	0.181
	Periphery	5	4	0.00	0.00	1.81 ± 0.52 (0.78–2.83)	
**BUB3 location**	Cytoplasm + nucleus	5	4	20.0	21.0	4.28 ± 2.36 (0.00–8.90)	0.567
	Nucleus	44	29	30.0	21.0	15.65 ± 4.01 (7.79–23.51)	
**SPINDLY extent**	0–49%	23	16	22.0	22.0	14.52 ± 5.37 (4.00–25.03)	0.294
	50–100%	26	16	34.6	27.7	12.26 ± 3.02 (6.33–18.20)	
**SPINDLY intensity**	Negative/weak/moderate	43	26	30.5	30.5	19.77 ± 4.33 (11.28–28.25)	0.060
	Strong	6	6	0.00	0.00	3.57 ± 2.04 (0.00–7.57)	
**SPINDLY distribution**	Homogeneous	12	9	15.3	0.00	5.15 ± 1.42 (2.37–7.94)	0.535
	Periphery/Multifocal	37	23	32.2	32.2	20.33 ± 4.63 (11.25–29.41)	

* Cumulative proportion (%) of survival time; ** mean for survival time in months (cancer-specific survival); ^†^ analysis performed for available biomarkers with follow-up information.

**Table 4 animals-12-03082-t004:** Multivariate analysis of the cancer-specific survival (CSS).

		Overall Survival	
Variables		HR (95% CI)	*p*-Value
**Histological type**	Papillary	1 (reference category)	
	Conventional	0.365 (0.026–5.117)	0.454
**Treatment**	Surgery	1 (reference category)	0.021
	Chemotherapy	2.326 (0.364–14.864)	0.372
	Palliative or support	6.214 (1.433–26.943)	0.015
**BUBR1 extent**	0, 1, 2+	1 (reference category)	
	3+, 4+	5.462 (1.455–20.502)	0.012
**Tumor stage**	I + II	1 (reference category)	
	III + IV	3.71 (0.909–15.135)	0.068
**Pattern of invasion**	I	1 (reference category)	0.200
	II	4.199 (0.863–20.416)	0.075
	III	5.910 (1.160–30.118)	0.033
	IV	6.587 (0.612–70.950)	0.120
**Stage of invasion**	I	-	-
	II	1 (reference category)	0.067
	III	2.192 (0.772–6.225)	0.141
	IV	15.686 (1.272–193.435)	0.032

HR, hazard ratio; CI, confidence interval for HR.

## Data Availability

The data information could be asked of from the corresponding author.

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
