# Peer review of "BUBR1 as a Prognostic Biomarker in Canine Oral Squamous Cell Carcinoma"

_animals, 2022, doi:10.3390/ani12223082_

Round 1

Reviewer 1 Report

The authors describe results of immunohistochemical analysis of several spindle assembly checkpoint components (BUBR1, BUB3 and SPINDLY proteins) in canine oral squamous cell carcinoma. In my oppinion, they submitted a high quality article presenting new results with promissing practical significance. The authors are aware of the relatively low number of analysed samples and the need of future research and confirmation of their results. Nevertheless, such works bringing into our attention possible new prognostic bimarkers and treatment targets are highly needed in veterinary medicine and animal cancer treatment.

My only minor comment to the authors:

Unify the usage of the used statistical test name to „Kaplan-Meier“ throughout the text (lines 258, 268)

Author Response

Response to Reviewer 1 Comments

Comments and Suggestions for Authors

The authors describe results of immunohistochemical analysis of several spindle assembly checkpoint components (BUBR1, BUB3 and SPINDLY proteins) in canine oral squamous cell carcinoma. In my oppinion, they submitted a high quality article presenting new results with promissing practical significance. The authors are aware of the relatively low number of analysed samples and the need of future research and confirmation of their results. Nevertheless, such works bringing into our attention possible new prognostic bimarkers and treatment targets are highly needed in veterinary medicine and animal cancer treatment.

RESPONSE: Thank you for your comments.

My only minor comment to the authors:

 Unify the usage of the used statistical test name to „Kaplan-Meier“ throughout the text (lines 258, 268)

RESPONSE: we corrected this now.

Thank you for all your comments.

Reviewer 2 Report

The paper provided by Delgado et al. refers to the immunohistochemical detection of crucial proteins controlling the cell division in canine oral squamous cell carcinoma, and the correlation of these protein levels with clinical and pathological parameters. The greatest value of the study is the inclusion of a series of strong clinical data of patients (e.g. survival time, metastasis presence, used treatments, etc.), and what follows their correlation with immunohistochemical staining results. The used methods are simple but are standard procedures in pathological research and diagnosis.
In the opinion of the reviewer, the submitted paper is important and brings new knowledge in the biology of canine oral squamous carcinomas and has a strong implementation potential in pathomorphological diagnostics.
Despite the appreciation for the authors' work and the very positive reception of the work, the reviewer would like to dispel some doubts:
•    In the reviewer’s opinion the first subunit of Results, 3.1 Patient characteristics should be included in the “Materials and Methods Section”
•    The reviewer would encourage authors to shift Table 3 and Table 5 to the Supplementary material because the correlation of survival with clinical and histopathological variables was not the goal of this study while the supplementary Fig. 1 refers to the main protein included in the title of the article (this table should be provided in the main manuscript).
•    In line 108, the authors wrote about non-specific binding blocking however no information about the blocking buffer was not provided
•    The authors used the mouse and rabbit anti-human antibodies for selected protein detection in canine tissues. Do the authors know about the cross-reactivity of the antibodies used? What is the concordance in the amino acid sequence of the studied proteins in humans and dogs? Can the authors provide such information?
•    The staining of canine or human tissues ware used as positive controls?
•    One of the most important conclusions is presented in figure 6, which is too small.
•    In the opinion of the reviewer, the authors in supplementary figures should not divide the groups according to the low and high immunostaining reaction extent/intensity (these data were shown in tables) and should show the exact values’ average (with Standard Error of the Mean (SEM)) of the immunohistochemical reaction for each group (e.g. breed size) and analyze using by another test eg. Tukey post hoc test. In the reviewer's assessment, this analysis is worth carrying out with all examined biomarkers and clinical and pathological variables. In current form descriptions of supplementary figures are very laconic.

Author Response

We provide the responses to the comments in the atached Word file.

Thank you for all comments.

Reviewer 3 Report

Dear Authors, your work is interesting and nicely presented. I just have some questions and corrections  for you: 

Row 51: it seems that the cited paper #2 does not give information about the role of tobacco and alcohol in Oral Squamous Cell Carcinomas in humans.  Cited paper #5 refers to the abuse of alcohol to have a role on the etiopathogenesis of this type of tumor, not the disuse of it. Please, adjust

Row 59: add ‘human’ OSCC, please

Row 71: write SAC signaling instead of spindle assembly checkpoint signaling

Row 79: write SAC component instead of spindle assembly checkpoint component

Row 87: I have a question, can we say that the guidelines of the Declaration of Helsinki were adapted to animal subjects? Can you comment that? I also would like to ask for adding a citation

Rows 89-91: did you consider as inclusion criteria a  minimum time for the available follow up? I think it is important for possible consequences to the statistical analysis. Probably you meant something about this referring to ‘valid information concerning survival’ at row 254

Row 92: I don’t think you should cite here the paper #20 instead to carefully expose the followed material and methods, even if they are part of another work on the same cohort of patients

Row 93: the cited paper #21 should not be cited in this section, you are just saying, indeed, that you collected age informations for statistical purposes

Row 110: can you add the clone number?

Row 170, figure 1: what about the intensity in figures c and d?

Row 224: even if well known, I would specify Ki-67 nuclear expression at the beginning of the sentence

Row 336: correct “significand

Check please that BUBR1 and BUB3 are written always correctly: you used both, upper and lower case letters (see: Table 2 and 5, rows 260, 268, 274, 317, 318, 344, 348)

Author Response

Comments and Suggestions for Authors

Dear Authors, your work is interesting and nicely presented. I just have some questions and corrections  for you: 

RESPONSE: Thank you for your comments.

Row 51: it seems that the cited paper #2 does not give information about the role of tobacco and alcohol in Oral Squamous Cell Carcinomas in humans.  Cited paper #5 refers to the abuse of alcohol to have a role on the etiopathogenesis of this type of tumor, not the disuse of it. Please, adjust

RESPONSE: We correct the reference 2 (we remove) as it was out of the sentence. Regarding the use of alcohol, we want to say the alcohol misuse. We corrected this in the text, and now the reference is logic in our opinion. Thank you for asking this.

Row 59: add ‘human’ OSCC, please

 RESPONSE: We corrected this now.

Row 71: write SAC signaling instead of spindle assembly checkpoint signaling

 RESPONSE: We corrected this now.

Row 79: write SAC component instead of spindle assembly checkpoint component

 RESPONSE: We corrected this now.

Row 87: I have a question, can we say that the guidelines of the Declaration of Helsinki were adapted to animal subjects? Can you comment that? I also would like to ask for adding a citation

 RESPONSE: The declaration of Helsinki is directed to humans research. However, there are some common principals and also reference to some animal research principals, for example, stating that “ the welfare of animals used for research must be respected”. Many of the principals of ethics of this guidelines were ethically adapted. However, we understand that this could be confusing and not directed to animal research, and we indicate now that ethical principals regarding animal research were followed (without mention the helsinky declaration). Please note that this was retrospective study and not having any experimentation on animals, so is not logic to include some references such as the european Directive 2010/63/EU as they is directed to lab experimental use of animals that was not the cope of our research.

Rows 89-91: did you consider as inclusion criteria a  minimum time for the available follow up? I think it is important for possible consequences to the statistical analysis. Probably you meant something about this referring to ‘valid information concerning survival’ at row 254

RESPONSE: We understand the comment of the reviewer. Survival analysis is a diffult evaluation in canine patients. In this way, we use the interval time of evaluation for a better analysis of survival as it includes not only the event as variable but also the interval of time for analysis. In the project we consider a minimum of one month, altought almost of the patients of the sample had a follow-up time superior than 6 months (unless the patient died) which is a good follow-up time for non-human studies. Nevertheless, and recognize that a long follow-up should be a goal, we include in limitation paragraph that the follow-up of the study could be higher and this could be a limition of the study.

Row 92: I don’t think you should cite here the paper #20 instead to carefully expose the followed material and methods, even if they are part of another work on the same cohort of patients

 RESPONSE: We agree with the author. We indicate also reference of classifications used when needed and also they appear in table one. For all these reasons we agree with the reviewer and deleted this citation.

Row 93: the cited paper #21 should not be cited in this section, you are just saying, indeed, that you collected age information’s for statistical purposes

RESPONSE: we correct this now.

Row 110: can you add the clone number?

RESPONSE: We include now this information;

“(clone 9, BD Biosciences, Sparks, MD, USA)”

Row 170, figure 1: what about the intensity in figures c and d?

RESPONSE: we correct this now. For these marker this was a moderate intensity that was classified as low expression intensity.

Row 224: even if well known, I would specify Ki-67 nuclear expression at the beginning of the sentence

 RESPONSE: we correct this.

Row 336: correct “significand”

 RESPONSE: we correct this.

Check please that BUBR1 and BUB3 are written always correctly: you used both, upper and lower case letters (see: Table 2 and 5, rows 260, 268, 274, 317, 318, 344, 348)

RESPONSE: we correct this.

Thank you for all your comments.

Reviewer 4 Report

The current manuscript entitled “BUBR1 as a prognostic biomarker in canine oral squamous cell 2 carcinoma 3” is a canine study similar to the author’s previous human publication "Expression of spindle assembly checkpoint proteins BubR1 and Mad2 expression as potential biomarkers of malignant transformation of oral leukoplakia: an observational cohort study”, also listed as reference #37 in the current study.  In the current study, the authors present SAC pathway expression data in a cohort of canine oral squamous cell carcinomas and tumors and correlate their findings with tumor pathology, grade, and patient survival.

My main concern/criticism with the current study is with the methods used to determine the survival analysis.

The authors write:  “Survival times of the patients who were still alive at the end of follow-up or who died from causes not associated with oral cancer were censured at the date they were last seen alive or at the date of death, respectively. We also consider dead by tumour the cases with loss of follow-up but with previous information of evidence of persistent/or advanced or disseminated disease or if they had died or been euthanised for reasons related to their tumour.”

The authors indicate that cases where death has not been confirmed (loss of follow up), but that had advanced disease at last follow up were considered to have died from the tumor, is that accurate?  If so, this is inconsistent with how patients without advanced disease and lost to follow up are scored. For patients that were without advanced disease at last follow up, but ultimately lost to follow up, the authors choose to censure those, but if a patient is lost to follow up, but was known to have progressive disease at last follow up, the authors choose to count those as “dead by tumor”.  For this Kaplan Meier analysis, the “event” is death, and if the event does not occur within the study time frame, either due to survival beyond the study date, or due to lack of follow up, ALL of those data points need to be censured, even if there is reason to believe the patient would have died soon after loss of follow up.  By not censoring those data points, the scoring method becomes subjective and biases any statistical analysis relating to survival.  Furthermore, how was the actual date of death determined for patients with advanced disease and lost to follow up?  How many patients fell into this category?

As such, survival and all related analysis correlating biomarker staining and categories listed in table 3 and 4 to survival needs to be redone with all patients lost to follow up correctly censured.

A supplementary table providing information on each patient that includes age (at diagnosis), breed, treatment, and cause of death or reason for censure (loss to follow up), and actual time interval from diagnosis to death is also warranted.

Additional General Comments:

1. Significant spelling and grammar editing is required.

2. Canine protein names should be in ALL CAPS (e.g. BURB1, CDC20).  Dogs and domestic species follow the same formatting rules as for humans.  Because this is a paper about canine oral cancers, it is assumed the authors are referring to canine proteins, unless otherwise stated.  For example, if the authors write “In mice, the MCC complex is composed of proteins Mad2, BubR1, and Bub3 in association with Cdc20 protein”, the lower case would be appropriate because they indicate they are referring to mouse proteins, not canine.  But if the authors are simply describing the protein pathways/interactions without naming a specific species, such as in lines 65 and 66, we assume they are canine proteins and they should be in ALL CAPS.

Specific Line Comments:

Introduction

1. Line 54-55 : “Cancer progression, namely OSCC development, is characterized by multiple genetic and epigenetic events.” – please add a reference to back up this statement.  

2. Suggestion: A figure showing an overview of the SAC and MCC complexes and interactions and how they influence mitosis checkpoints would be helpful.  The relationship of SPINDLY and dynein to MAD2, BABR1, and BUB3 in the context of the SAC is not clear.

3. Line 70 – replace “Other” with “Another”

4. Lines 148-149:  For univariate survival analysis, Kaplan–Meier and log-rank tests were performed.  For multivariable analysis, Cox proportional hazards model was used (enter method), considering variables with significant influence in the univariate analyses.”  I am not familiar with the “enter method” of multivariable analysis… or did someone just forget to add information regarding the methodology here?   

 5. Tables 3 and 4 – the numbers for some rows and columns are inconsistent with each other and the results.

6. The CSS Mean is in months, correct?  Please indicate that in the table caption.  I see it is noted the methods section, but it should be indicated in the table also.

7. Table 3:

Tumor Stage (1st row):  I (n=6), II (n=9), III (n=12), IV=(n=17) ; I and II (n=15), II and IV (n=29)

Tumor Stage (2nd row): I and II (n=14), III and IV (n=30)

One patient jumped from low grade to high grade in the second combined analysis. Please explain.   Also, there are only 44 samples included here for tumor stage.  You note in Table 1 that there were only 50/60 samples available for some analysis, but there is no mention of that situation in table 3.

8. Table 3 – Treatment column – total dead = 43.  That's 10 more than the other rows reported.

9. Tables 3 and 4 – What is shown in the columns for “CSS 1-year and 2-year”?  It appears to be the percent of cases that survived, is that accurate?  Please include this detail in the table.

10. Table 4 – the N for each row group changes.

BUBR1 extent (n=50), BUBR1 intensity (n=50), BURBR1 location (n=49)  What happened to the other sample for “BUBR1 location” row?  Then there are only 49 samples in each row after that.

Author Response

Dear reviewer 

Please find attached the word file with the responses to the comments.

Thank you for all the comments.

Round 2

Reviewer 4 Report

I am confused by the authors’ response regarding my concerns with the methodology used to determine CSS, specifically about including patients known to be terminally ill, but lost to follow up in the “death by tumor” event group, rather than as censored data.  In one paragraph, the authors provide further argument for why they chose to use these methods, even when the date of death was not known due to loss of follow up.  In the next paragraph of their response, however, the authors acknowledge that, by the end of the study, they did have actual dates of death for all dogs and there were no “patients without a dead date in a group of events”. Can the authors clarify this?

If that is the accurate, then only cases where death was not associated with oral cancer would be censored.  However, the methods still describe “Survival times of the patients who were still alive at the end of follow-up or who died from causes not associated with oral cancer were censured at the date they were last seen alive or at the date of death, respectively” – meaning that some dogs were lost to follow up and therefore the dates of death were not known.  Can the authors clarify these conflicting statements?

The authors have edited the methods section to remove the language describing dogs that were lost to follow up, but had terminal illness at last follow up. The text from version 1 of the manuscript: “We also consider dead by tumour the cases with loss of follow-up but with previous information of evidence of persistent/or advanced or disseminated disease or if they had died or been euthanised for reasons related to their tumour.”

Assuming that some dogs were lost to follow up, the new language implies that all dogs who were “still alive at the end of follow up”, regardless of disease state, “were censored at the date they were last seen alive”.  Can the authors confirm this?

If a date of death was known for all patients by the end of study, please include a sentence in the methods that states this.

However, if for some cases the authors could “somehow” confirm the patient had died (by tumor), but they did not have an exact date of death, what was used for the “time of event” for those cases?  The authors would need to elaborate on the methods used to determine the date of death event assigned for those cases.  For example, did the authors use the date at which they were informed the patient had died (having died sometime prior to this date)?  Or did they use the date of last follow up with the assumption the patient died shortly after?  That being said, if the exact date of death was known for these cases and the authors added that statement to the methods, then this comment is already resolved.

The next questions are regarding Section 3.6. Clinical outcome:

Manuscript text: “Fifty patients had valid information concerning survival at the end of the study (mean follow-up of 6.8±1.5 months), of which 5 (10%) were alive without oral cancer, 6 (12%) were alive with oral cancer, 33 patients (66%) had died as a result of oral cancer and 6 (12%) died from other causes.”

Can the authors define “valid information concerning survival” in this section.  Are the authors saying that patients who were lost to follow up were considered to have “valid” survival information, or are they saying no patients were lost to follow up?

What is the “mean follow up of 6.8±1.5 months” describing here?  Is that the mean “last follow up” since time of diagnosis?  Please clarify. 

The authors state that, by the end of study, “5 (10%) were alive without oral cancer and 6 (12%) were alive with oral cancer” and that the rest (39) had died either of oral cancer (33) or other causes (6).  This implies that there were 11 patients confirmed to be alive at the end of the study and were not lost to follow up, is that accurate?  If some of these patients were, in fact, lost to follow up, please indicate the number of patients censored due to loss to follow up in each group and adjust the language to indicate which patients were alive “at last follow up” and which were alive “at the end of study”.  This is a significant distinction.

For example, the authors could write “at the end of the study, XX patients were lost to follow up (censored), XX patients were alive without oral cancer, XX patients were alive with oral cancer, XX patients died due to oral cancer, and XX patients died from other causes (censored).    Of those lost to follow up, XX were alive with cancer and XX were alive without cancer at last follow up.”

Specific Line edits/comments:

Table 1: Pattern and Stage of Invasion rows both have stage “II” written twice, rather than II and III.

Lines 343-345:  “In previous study of human OSCC…”  change to “In a previous study…” 

Also, there is no citation for the study referenced here.

Line 372 – “We could also expect that these differences are also related to species differences.”

It is not clear which species-related differences the authors are referring to here.  The current results in canine OSCC match those in human OSCC studies; they are not different.  The author’s comment regarding tumor type/location differences is supported by the different results from studies using different tumor types, but within the same tumor type (OSCC) I do not see any conflicting data between species (based on the limited available studies).

All my other comments and concerns have been appropriately addressed by the authors.  However, clarification regarding the methods described above is needed prior to publication, especially given the conflicting statements by the authors.

Author Response

Comments and Suggestions for Authors

I am confused by the authors’ response regarding my concerns with the methodology used to determine CSS, specifically about including patients known to be terminally ill, but lost to follow up in the “death by tumor” event group, rather than as censored data.  In one paragraph, the authors provide further argument for why they chose to use these methods, even when the date of death was not known due to loss of follow up.  In the next paragraph of their response, however, the authors acknowledge that, by the end of the study, they did have actual dates of death for all dogs and there were no “patients without a dead date in a group of events”. Can the authors clarify this?

RESPONSE: As we write before, when we design the project before performing the study, we considered that in case that we know ”somehow” that the patients (with terminal disease – persistent/advanced/disseminated) has died by tumour (for instance at home but the owner could not give a precise date of dead after last vet visit) it would be considered an event. During the study, until the end we could confirm in every patient the date of dead. Please note that we clarify this in the paper and this potential issue does not put now in the paper.

If that is the accurate, then only cases where death was not associated with oral cancer would be censored.  However, the methods still describe “Survival times of the patients who were still alive at the end of follow-up or who died from causes not associated with oral cancer were censured at the date they were last seen alive or at the date of death, respectively” – meaning that some dogs were lost to follow up and therefore the dates of death were not known.  Can the authors clarify these conflicting statements?

RESPONSE: Please note that we correct this as probably there were some misunderstanding on the term used. This is an temporal evaluation and the study consider what is happening during of time of follow-up and then the event (dead by tumour). So we have the date of dead by tumour (event) and for the other patients the date of last visit alive (they are not dead by tumour) (this could include patients without cancer or with cancer disease that are not dead and we don’t know what will happen next) or dead for other causes that not cancer. Please note that this is Cancer-specifc survival and not a disease-fre survival. We are analysing only the cases of dead by tumour. So here we don’t see bias regarding the this methodology uses (Using Kaplan-Meire curves). Perhaps the sentence could be misunderstood and we change this now for a better and objetive way. Probably the word “censored”  and the expresion “ lost of follow-up” could be confuse in this context.  When we say “lost of follow-up is not without follow-up. For an example, there are cases that the follow in schedule for 5 years but the patient disapear after 3 years. In this cases the KM methodology indicates to use the time between the considered initial point date and the last visit and the state of the patients that could be an time of 24 months and a state of alive without disease (for example). So we are not excluding this patient only consider the time that we have information.  We perform multiple research on this and probably this issue is related with confusion on writing the sentence. We simplify in the paper to give the maximum objetive to this. We hope that now this is understandable. Thank you for increase the clarity in the sentence.

“Survival times of the patients who were still alive at the end of follow-up or who died from causes not associated with oral cancer were consider using the last date that they were seen alive or at the date of death, respectively.“

The authors have edited the methods section to remove the language describing dogs that were lost to follow up, but had terminal illness at last follow up. The text from version 1 of the manuscript: “We also consider dead by tumour the cases with loss of follow-up but with previous information of evidence of persistent/or advanced or disseminated disease or if they had died or been euthanised for reasons related to their tumour.”

Assuming that some dogs were lost to follow up, the new language implies that all dogs who were “still alive at the end of follow up”, regardless of disease state, “were censored at the date they were last seen alive”.  Can the authors confirm this?

RESPONSE: We include the last date that the dog was seen and the state of the dog. If there is a dead, here this date is substituted by the dead date as the methodology principals of this KM evaluation indicate. Also please note that when we say “some dogs were lost to follow up, ” it means that they have follow-up but then they disapear from the following appointment. This is different of not having follow-up of the cases (these ones were excluded from this analysis). We also explain this before and change the grammar in the text to be more clear.

If a date of death was known for all patients by the end of study, please include a sentence in the methods that states this.

RESPONSE: We include now this

However, if for some cases the authors could “somehow” confirm the patient had died (by tumor), but they did not have an exact date of death, what was used for the “time of event” for those cases?  The authors would need to elaborate on the methods used to determine the date of death event assigned for those cases.  For example, did the authors use the date at which they were informed the patient had died (having died sometime prior to this date)?  Or did they use the date of last follow up with the assumption the patient died shortly after?  That being said, if the exact date of death was known for these cases and the authors added that statement to the methods, then this comment is already resolved.

RESPONSE: Please note that we already correct this as we have all dead date for the patients. We also have corrected this in the article and include now the sentence asked by the reviewer.

The next questions are regarding Section 3.6. Clinical outcome:

Manuscript text: “Fifty patients had valid information concerning survival at the end of the study (mean follow-up of 6.8±1.5 months), of which 5 (10%) were alive without oral cancer, 6 (12%) were alive with oral cancer, 33 patients (66%) had died as a result of oral cancer and 6 (12%) died from other causes.”

Can the authors define “valid information concerning survival” in this section.  Are the authors saying that patients who were lost to follow up were considered to have “valid” survival information, or are they saying no patients were lost to follow up?

REPONSE: We agree that this word “valid”is confusing and we eliminate now this. All information on methods are detailed in methods section. The cases that had no follow-up information were eliminate.

What is the “mean follow up of 6.8±1.5 months” describing here?  Is that the mean “last follow up” since time of diagnosis?  Please clarify.

Response: We explain this in the methods. It is since histologic diagnosis to last date consider according the methods.

The authors state that, by the end of study, “5 (10%) were alive without oral cancer and 6 (12%) were alive with oral cancer” and that the rest (39) had died either of oral cancer (33) or other causes (6).  This implies that there were 11 patients confirmed to be alive at the end of the study and were not lost to follow up, is that accurate?  If some of these patients were, in fact, lost to follow up, please indicate the number of patients censored due to loss to follow up in each group and adjust the language to indicate which patients were alive “at last follow up” and which were alive “at the end of study”.  This is a significant distinction. For example, the authors could write “at the end of the study, XX patients were lost to follow up (censored), XX patients were alive without oral cancer, XX patients were alive with oral cancer, XX patients died due to oral cancer, and XX patients died from other causes (censored).    Of those lost to follow up, XX were alive with cancer and XX were alive without cancer at last follow up.”

RESPONSE: Please note that this is a interval time analysis as peformed by Kaplan-meier curves. This is not a chi-test analysis of patients alive or dead at 3, 4, .. years. Again probably the term “censored” is misundertood and we correct this is the text. All patients with information regarding follow-up were included  (50). Of this 5 (10%) were alive without oral cancer and 6 (12%) were alive with oral cancer” and the rest (39) had died either of oral cancer (33) or other causes (6). 

When we said “lost of follow-up”, we are saying that the patients was performing regular appointments during example 2 years by then it disappear from the clinical. Please note that we have here the follow-up and the state of the dog on last follow-up visit that is an accurated information regarding the use of KM method. We don’t want to say that there is no follow up. We correct this grammar as perhaps this is leading to some confusion. Also when we say “end of study” it mean that when we perfom all analysis. We are not refer to any follow time in particluar. We also correct this as this could also be misundertanding. Thank you for the comment.

Specific Line edits/comments:

Table 1: Pattern and Stage of Invasion rows both have stage “II” written twice, rather than II and III.

RESPONSE: We correct this now. Thank you for ask this.

 Lines 343-345:  “In previous study of human OSCC…”  change to “In a previous study…”

RESPONSE: We correct this now.

Also, there is no citation for the study referenced here.

RESPONSE: We correct this now. Thank you

Line 372 – “We could also expect that these differences are also related to species differences.”

It is not clear which species-related differences the authors are referring to here.  The current results in canine OSCC match those in human OSCC studies; they are not different.  The author’s comment regarding tumor type/location differences is supported by the different results from studies using different tumor types, but within the same tumor type (OSCC) I do not see any conflicting data between species (based on the limited available studies).

RESPONSE: We agree with authors and eliminate this.

All my other comments and concerns have been appropriately addressed by the authors.  However, clarification regarding the methods described above is needed prior to publication, especially given the conflicting statements by the authors.

 We thank for the comments of the reviewer and hope that’s satisfy the comments. Thank you for every comment.